# Lysine Deprivation during Maternal Consumption of Low-Protein Diets Could Adversely Affect Early Embryo Development and Health in Adulthood

**DOI:** 10.3390/ijerph17155462

**Published:** 2020-07-29

**Authors:** Lon J. Van Winkle, Vasiliy Galat, Philip M. Iannaccone

**Affiliations:** 1Department of Biochemistry, Midwestern University, Downers Grove, IL 60515, USA; 2Department of Medical Humanities, Rocky Vista University, 8401 S. Chambers Road, Parker, CO 80134, USA; 3Department of Pathology, Northwestern University Feinberg School of Medicine, Stanley Manne Children’s Research Institute and the Ann and Robert H. Lurie Children’s Hospital of Chicago, Chicago, IL 60209, USA; v-galat@northwestern.edu; 4Departments of Pediatrics and Pathology, Northwestern University Feinberg School of Medicine and the Lurie Children’s Hospital of Chicago, Chicago, IL 60209, USA; pmi@northwestern.edu

**Keywords:** barker hypothesis, blastocyst, embryo, embryonic stem cells, glutamate signaling, low-protein diet, lysine, offspring

## Abstract

The conversion of lysine to glutamate is needed for signaling in all plants and animals. In mouse embryonic stem (mES) cells, and probably their progenitors, endogenous glutamate production and signaling help maintain cellular pluripotency and proliferation, although the source of glutamate is yet to be determined. If the source of glutamate is lysine, then lysine deprivation caused by maternal low-protein diets could alter early embryo development and, consequently, the health of the offspring in adulthood. For these reasons, we measured three pertinent variables in human embryonic stem (hES) cells as a model for the inner cell masses of human blastocysts. We found that RNA encoding the alpha-aminoadipic semialdehyde synthase enzyme, which regulates glutamate production from lysine, was highly expressed in hES cells. Moreover, the mean amount of lysine consumed by hES cells was 50% greater than the mean amount of glutamate they produced, indicating that lysine is likely converted to glutamate in these cells. Finally, hES cells expressed RNA encoding at least two glutamate receptors. Since this may also be the case for hES progenitor cells in blastocysts, further studies are warranted to verify the presence of this signaling process in hES cells and to determine whether lysine deprivation alters early mammalian embryo development.

## 1. Introduction

As shown by in vitro studies, amino acids derived from dietary protein are likely to support normal preimplantation embryo development in vivo [1]. For example, a five-minute exposure of fertilized mouse eggs to a medium without added amino acids impairs their development [2]. Essential amino acids, including lysine, promote development of blastocysts with more cells in their inner cell masses by increasing the cleavage rate of embryos. Such blastocysts give rise to viable fetuses upon transfer to surrogate mothers more frequently than blastocysts that develop in vitro without essential amino acids [3]. Blastocysts developing in a medium containing essential amino acids also produce larger fetuses. While essential amino acids support growth and development as nutrients, they also foster signaling in early embryos [1,4,5,6]. Amino acid transport proteins are needed in plasma membranes to regulate the concentrations of essential amino acids in conceptuses, but these transporters can be overwhelmed by altered amino acid availability.

For example, the concentrations of the essential cationic amino acids-lysine, arginine, and histidine-are greatly decreased in blastocysts when the culture medium contains a physiological concentration of the zwitterionic amino acid methionine, but no other amino acids [7]. A 30-fold increase in amino acid transport system b^0,+^ activity between the eight-cell and blastocyst stages of embryo development accounts for the effect of methionine on the concentrations of other amino acids in blastocysts [8]. System b^0,+^ functions primarily to exchange zwitterionic for cationic amino acids under physiological conditions [9]. Interestingly, maternal consumption of a low-protein diet (LPD) results, unexpectedly, in an increase in the uterine fluid methionine concentration on day four of pregnancy, while the concentration of lysine in blastocysts decreases [10].

Maternal LPDs adversely affect peri-implantation blastocyst development and the subsequent growth of fetuses and offspring [5,6,11]. These offspring have increased risk of exhibiting metabolic syndrome and related disorders as adults. It is likely that the decreased amino acid availability caused by maternal LPDs alters how stem cell progenitors function in these early embryos. For example, in the mouse embryonic stem (mES) cell model, threonine deprivation causes the cells to lose their pluripotency and stop proliferating [12]. Similarly, methionine deprivation halts human embryonic stem (hES) cell proliferation [13]. While LPDs are low in threonine and methionine, they are also low in other essential amino acids, including lysine.

Lysine deprivation blocks hES cell proliferation almost completely [13]; therefore, LPDs could adversely affect blastocyst development through lysine deprivation of hES progenitor cells in their inner cell masses. Withholding several other essential amino acids from hES cells does not impair their proliferation [13], so the effect of lysine deprivation on hES cells is unlikely due to lysine’s essential nature. Lysine is, however, catabolized to glutamate in mouse and human brains, and this metabolism appears likely to be required for normal brain functioning through needed glutamate signaling [14,15,16,17]. Although the glutamate–glutamine cycle functions as an established source of glutamate in the mammalian brain [18], this cycle seems insufficient for brain health [14,15,16,17,18].

Similarly, metabotropic glutamate receptor signaling by endogenously produced glutamate helps to maintain undifferentiated, pluripotent mES cells in culture [19,20] and likely helps to maintain mES progenitor cells in peri-implantation embryos. Nevertheless, the mechanism of glutamate production in mES cells remains to be determined. Glutamine could conceivably be converted to glutamate by glutaminase in mES cells, since mES cell culture media have historically included an abundant supply of glutamine [19,20]. However, most hES cell culture media contain a “stable form of glutamine”, such as alanyl-glutamine [21,22]. The metabolism of these stable forms of glutamine in hES cells has not, to our knowledge, been studied. For these reasons, we asked the following questions:Is RNA, encoding the protein that regulates glutamate synthesis from lysine (alpha-aminoadipic semialdehyde synthase), relatively abundant in hES cells?Is the amount of lysine consumed by hES cells equal to or greater than the amount of glutamate the cells produce?Is RNA encoding one or more metabotropic glutamate receptors expressed in hES cells?

## 2. Materials and Methods

### 2.1. hES Cell Culture

Six hES cell lines-H1 (WA01), H7 (WA07), H9 (WA09), and H14 (WA14) (WiCell, Madison, WI, USA) and CM7 and CM14 (established at Galat lab [23])-were used for this study. For the microarray analysis H7, H9, H14, CM7, and CM14 cells were grown in StemPro medium (Invitrogen, Carlsbad, CA, USA) on a Matrigel substrate (BD Bioscience, San Jose, CA, USA). The confluent cultures of hESCs growing on 10-cm dishes were split to 6 experimental 10-cm dishes mechanically by using the StemPro EZ Passage tool (Invitrogen). For amino acid assays, H9 cells were maintained in DMEM/F12 and supplemented with Knockout Serum Replacement (SR-1), FGF, 2,β-mercaptoethanol, and GlutaMax, Gibco Inc., Billings, MT, USA). For RNAseq analysis, H1 cells were maintained on Matrigel-coated culture dishes in StemMACS iPS-Brew XF (Miltenyi Biotec, Bergisch Gladbach, Germany). A single-cell suspension was counted with the help of an automatic cell counter, Nexcelom (SelectScience, Church Farm Business Park, UK), during cell lifting for the analysis. Cell confluence and morphology were observed daily.

### 2.2. Mesenchymal Differentiation

Mesenchymal cells (MSC) were established as previously described [24]. Briefly, mesendodermal induction of H1 hES cells was facilitated by OP9 stromal cells. The isolated APLNR+ progenitors were plated as single cells in semisolid colony-forming serum-free medium (CFSFM). After 2 weeks, the mesenchymal colony-forming units (MS-CFU) were manually picked, and the MSC cells were transferred to an adherent culture and maintained in EGM-2 medium (Lonza, Basel, Switzerland).

### 2.3. Osteogenic Differentiation

The mesenchymal cells described above were further subjected to osteogenic differentiation which was induced using the MSCgo Osteogenic Differentiation Medium (Biological Industries, Beit HaEmek, Israel) according to manufacturer’s instructions.

### 2.4. Vascular Differentiation

The endothelial differentiation of H1 hES cells was established by a monolayer induction protocol as previously described [25]. Briefly, single cells were plated on 60-mm culture dishes coated with Matrigel and cultured overnight in StemMACS iPS-Brew XF (Miltenyi Biotec). Differentiation was induced with an induction media containing CHIR990921 (6 μM) added on day 0. On day 2 of induction, CHIR990921 was removed from the media. The cells were collected on day 5 of differentiation.

### 2.5. Neural Differentiation

The differentiation of H1 hES cells to neural progenitor cells was accomplished as previously described [26]. Briefly, the neuronal rosettes were mechanically detached from the differentiated hES cell colonies and plated on poly-ornithine/laminin-coated tissue culture dishes in the ENStem-A neural expansion medium containing 20 ng/mL of EGF/FGF2. Acutase (MilliporeSigma, Burlington, MA, USA) was employed for cell detachment and passaging.

### 2.6. RNA Isolation

For RNA analysis, confluent cultures were lifted by using 0.05% trypsin (Cellgro, Lincoln, NE, USA), washed in ice-cold phosphate-buffered saline (PBS, Arlington, VA, USA), and pellet-stored at −80 °C. Total mRNA isolation from the cell lines for the microarrays was performed with the PureZOL RNA isolation reagent (Bio-Rad, Hercules, CA, USA), according to the manufacturer’s instructions. Total RNA for RNAseq was extracted with the RNeasy Mini Kit (Qiagen, Hilden, Germany) via the instructions provided in the manufacturer’s protocol. RNA quality and concentration were assessed with Nanodrop instrument ND-1000 Spectrophotometer (Thermo Fisher Scientific, Waltham, MA, USA).

### 2.7. Microarray Data Preparation

RNA aliquots were submitted to University of Chicago Genomics Facility. The RNA samples were reverse-transcribed into cDNA, which was hybridized onto a HumanHT-12 v4 BeadChip that was scanned by Illumina iScan. The acquired data were processed and normalized via the iScan Control software. Fold-change gene expression comparisons were obtained using R Studio software (Bioconductor package). Another dataset was established on NimbleGen array Platform: 47,634 tags (human). Images were processed in NimbleGen software, resulting in quantile-normalized signal intensities for each probe. Therefore, no additional preprocessing and normalization procedures were applied to the data.

### 2.8. RNA Sequencing Analysis

RNASeq expression data were obtained for the H1 hES cell line. Aliquots of RNA were submitted to Northwestern University’s NUSeq Core. The mRNA library was prepared and the samples were analyzed using HiSeq 4000 Sequencing (50 bp, single reads). The list of differentially expressed genes obtained was further analyzed using MetaCore (version 20.1.1, Clarivate Analytics, Philadelphia, PA, USA) and R Studio software (version 3.6.1, Boston, MA, USA).

### 2.9. Amino Acid Production/Utilization

To measure changes in the amino acid concentrations in culture medium, hES cell colonies (H9 cell line) were grown on mouse feeder cells. The amino acid concentrations in the medium were determined daily for 4 days of culture and before changing the medium, using HPLC as described previously [4,27]. These concentrations were converted to percentages of the initial amino acid concentrations in the medium and compared to percentage amino acid concentration changes in cultures containing either feeder cells alone or no cells.

### 2.10. Statistical Analyses

Statistical analyses were performed using GraphPad Prism 8.0.2 software (GraphPad Software Inc., La Jolla, California, CA, USA). We used one-sample *t*-tests to determine whether or not mean values differed significantly from zero; furthermore, mean values were also compared statistically using analysis of variance (ANOVA) in combination with multiple comparison tests and *t*-tests. ROUT (Q = 1%) was used to determine whether any values in each set of data were statistically significant outliers. This study was reviewed and found to fulfill the criteria for exemption by the Midwestern University and Northwestern University Institutional Review Boards (IRBs).

## 3. Results

### 3.1. Expression of RNA Encoding Alpha-Aminoadipic Semialdehyde Synthase (LKRSDH)

RNA encoding alpha-aminoadipic semialdehyde synthase was relatively abundant in hES cells (Figure 1). The mean level exceeded that of the two housekeeping genes RPL41 and SIRT6 by more than 3-fold and more than 23-fold (*p* < 0.0001), respectively. Moreover, alpha-aminoadipic semialdehyde synthase RNASeq expression decreased to 1/22 of its level in H1 hES cells as these cells differentiated into mesenchymal cells (*p* < 0.0001), and a similar effect was found for osteogenic, vascular, and neural differentiation (Figure 1; ANOVA, Effect Size = *r* = 0.999, crucial practical importance, *p* < 0.0001). Alpha-aminoadipic semialdehyde synthase RNA was also 14 times more abundant than glutaminase 1 and 2 reads in hES cells (Figure 1). Alpha-aminoadipic semialdehyde synthase regulates production of glutamate from lysine, whereas glutaminase converts glutamine to glutamate [14,15,16,17,18].

### 3.2. Production/Utilization of Glutamate and Lysine by hES Cells (H9 Cell Line)

Human ES cells consumed a mean of 9.0% of the lysine in the medium relative to feeder cells alone (Figure 2a)-a change in the lysine concentration of 0.045 mM (0.09 × 0.50 mM, the concentration of lysine in the medium). In contrast, hES cells increased the concentration of glutamate in the medium by a mean of 62% compared to feeder cells alone (Figure 2b), which is an increase of 0.031 mM (0.62 × 0.050 mM, the concentration of glutamate in the medium). Endogenously produced glutamate helps to maintain pluripotency and proliferation of mES cells through metabotropic glutamate receptor 5 signaling [19,20].

### 3.3. hES Cells Express at Least Two Metabotropic Glutamate Receptors

RNAseq established that hES cells express metabotropic glutamate receptor 3 as well as 5 (Figure 1). While expression of the GRM5 gene was very low and did not register using the scale in Figure 1, its mean expression was greater than zero (one-sample *t*-test, *p* = 0.01). Assuming that the receptor proteins are expressed by hES cells, glutamate in the medium (or produced endogenously by their progenitor cells in peri-implantation blastocysts) should activate glutamate receptor signaling [14,15,16,17,18,19,20].

## 4. Discussion

In the major pathway for its catabolism, lysine is first converted to alpha-aminoadipic semialdehyde and glutamate by alpha-aminoadipic semialdehyde synthase [14,15,16]. Human fibroblasts, astrocytes, and neural progenitor cells each express this protein [16]. The pool of glutamate produced in this way must be unique in comparison to glutamate formation from other amino acids. Human alpha-aminoadipic semialdehyde synthase deficiencies are associated with severe physiological and morphological defects in the nervous system [14]. Moreover, deficiencies of the next enzyme in the lysine degradation pathway, antiquitin, are the major cause of vitamin B6-dependent epilepsy [16]. Similarly, diminished lysine catabolism to glutamate in other neurological disorders appears to cause the synaptic dysfunction in Opa1-deficient patients [28]. More broadly, plants as well as animals depend on lysine catabolism to foster glutamate signaling [29]. Conversion of lysine to glutamate for metabotropic glutamate receptor signaling in both mES and hES cells (Figure 1 and Figure 2) likely fosters their pluripotency and proliferation [19,20].

As for the removal of lysine from an ES cell culture medium [13], LPDs limit the supply of lysine to ES progenitor cells in blastocysts. The lysine concentration in mouse blastocysts is diminished by a maternal LPD [10], but the mechanism of this diminution is likely not caused by the concomitant decreases of lysine levels in blood or uterine fluid. We propose that the blastocyst trophectoderm poses a barrier to delivery of amino acids to the inner cell mass, much like endothelial cells in the central nervous system form the blood–brain barrier. In the case of lysine, its transporters in the trophectoderm plasma membrane are highly selective of other amino acids over lysine [1,5]. Unlike the blood–brain barrier, however, the trophectoderm exhibits high endocytic activity [5,6,11]. The mouse trophectoderm appears to take up proteins in this manner and to hydrolyze them to supply threonine to the inner cell mass [5]. This could be the mechanism by which the mouse trophectoderm supplies lysine. Moreover, the concentration of methionine in uterine secretions increases in concomitance with maternal consumption of an LPD [10], and lysine can be lost from blastocysts in exchange for methionine via amino acid transport system b^0,+^ [1,7].

Much remains to be learned about how various amino acids are provided to the inner cell mass and processed by these cells. This processing seems to include specialized mitochondria that compartmentalize amino acid metabolism for specific purposes [6]. In the present case, such mitochondria would take up lysine specifically for glutamate production and its resultant signaling among hES progenitor cells via metabotropic glutamate receptors. Similarly, in the brain, glutamate production from lysine must function in a manner distinct from its production by other amino acids such as glutamine. Impaired glutamate production from lysine can cause numerous abnormalities in the central nervous systems of mammals, including humans [14,15,16,28].

## 5. Limitations/Future Directions

We studied gene expression at the RNA level rather than the protein level, and we indirectly examined the possible conversion of lysine to glutamate through changes in the concentrations of these amino acids in an hES cell culture medium. Hence, catabolism of lysine to glutamate and the resultant signaling via metabotropic glutamate receptors remain to be examined using ^15^N-labeled lysine and the direct effects of glutamate on hES cell proliferation, respectively. Nevertheless, the developmental origins of health and disease (Barker) hypothesis applies well to all mammalian species [5,6,11], and a maternal LPD has been shown to cause less lysine to remain in mouse blastocysts and their inner cell masses [10]. We present evidence for one process by which impaired lysine delivery to ES progenitor cells could influence their development and differentiation. The existence of this mechanism should be studied further.

## 6. Conclusions

All three of our research questions were affirmed. Using RNAseq, we found the enzyme regulating lysine catabolism to glutamate to be highly expressed in hES cells. In addition, more than enough lysine was consumed by hES cells in culture to account for all of the glutamate the cells produced. Finally, hES cells were found to express at least two metabotropic glutamate receptors. These results for the hES cell model of blastocyst inner cell masses (ICMs) support the conclusion that glutamate signaling among cells in ICMs helps to maintain their pluripotency and proliferation. Hence, maternal consumption of LPDs could impair delivery of lysine for glutamate production and signaling among ES progenitor cells in blastocysts and thus alter early embryo development and the health of resultant offspring.

## Figures and Tables

**Figure 1 ijerph-17-05462-f001:**
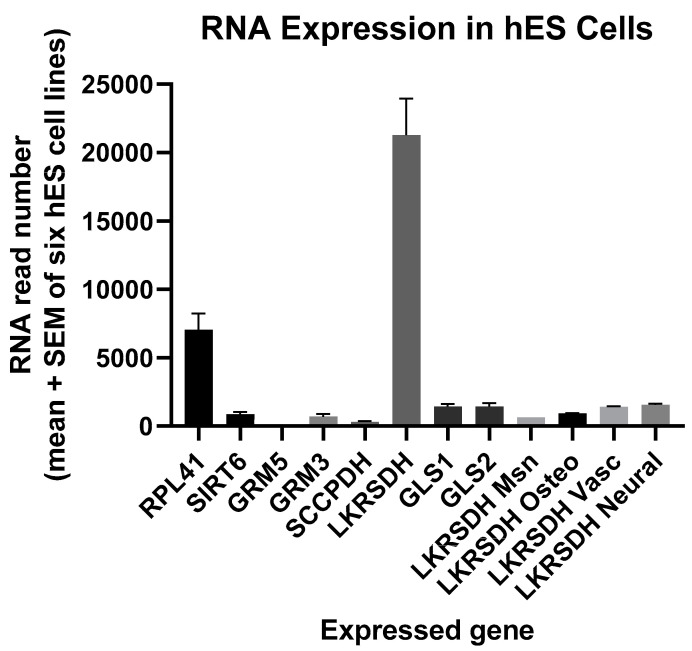
The mean RNA read number for alpha-aminoadipic semialdehyde synthase (LKRSDH) in human embryonic stem (hES) cells far exceeded other read numbers examined, but such was the case only before hES cells differentiated into mesenchymal (Msn), osteogenic (Osteo), vascular (Vasc), or neural cells (microarray plus RNASeq data, *n* = 2–6 per group, ANOVA, *p* < 0.0001). All mean read numbers were statistically significant and greater than zero (*p* = 0.02 or less for each of the mean read numbers). RPL41, 60S ribosomal protein L41; SIRT6, sirtuin 6 (stress-responsive protein deacetylase and mono-ADP ribosyltransferase); GRM5, glutamate metabotropic receptor 5; GRM3, glutamate metabotropic receptor 3; SCCPDH, saccharopine dehydrogenase (putative); LKRSDH, alpha-aminoadipic semialdehyde synthase; GLS1, glutaminase 1; GLS2, glutaminase 2; LKRSDH Msn, LKRSDH in mesenchymal cells formed from H1 hES cells; LKRSDH Osteo, LKRSDH in osteogenic cells formed from H1 hES cells; LKRSDH Vasc, LKRSDH in vascular cells formed from H1 hES cells; LKRSDH Neural, LKRSDH in neural cells formed from H1 hES cells.

**Figure 2 ijerph-17-05462-f002:**
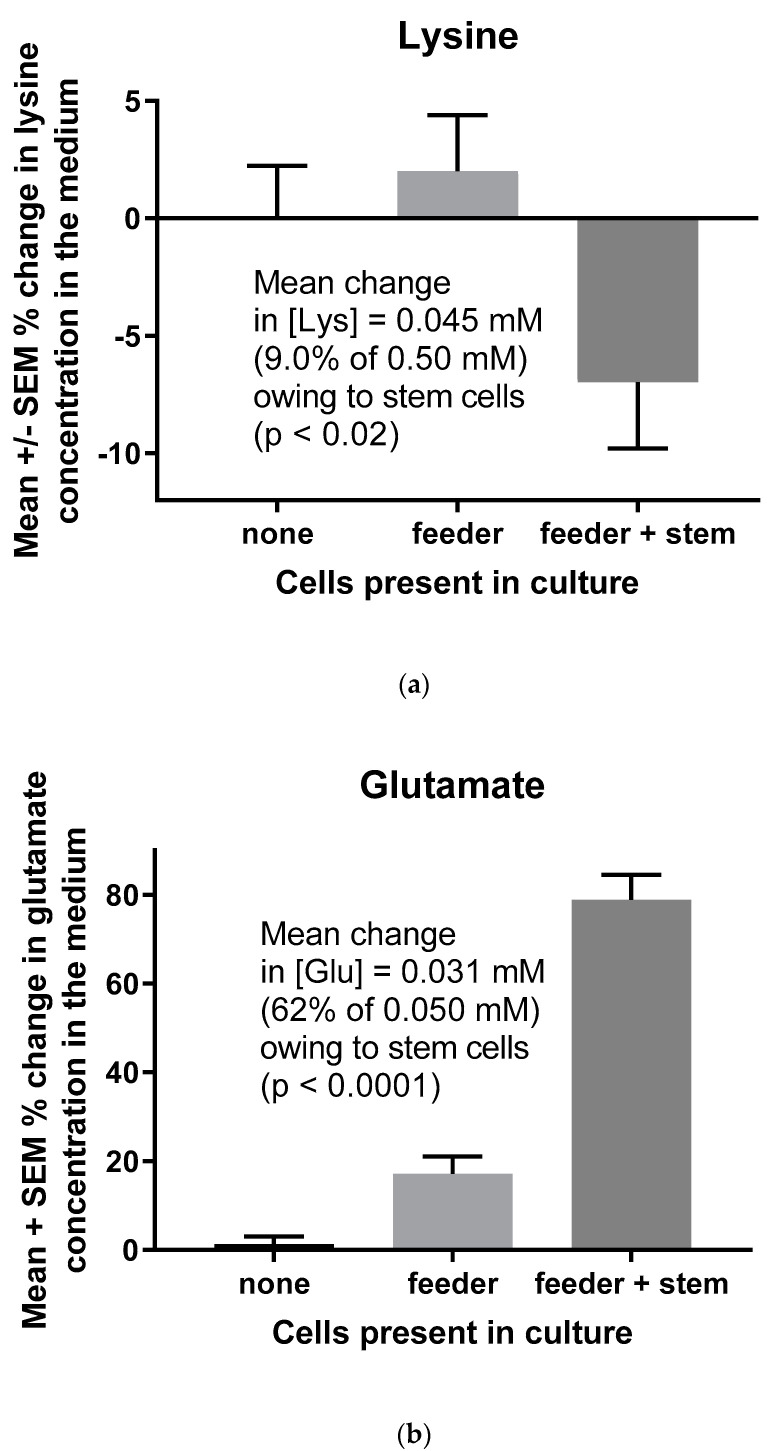
Lysine consumption (**a**) and glutamate production (**b**) by hES cells (H9 cell line, WA09). Cell-free control cultures were performed to show that, in the absence of cells, lysine and glutamate concentrations were unchanged in the medium. Lysine consumption and glutamate production were the differences between mean values in the presence of feeder cells vs. feeder plus stem cells (*n* = 34 determinations of the lysine concentrations in the medium and *n* = 36 determinations of the glutamate concentrations in the medium). The *p* values are for *t*-tests between feeder cells vs. feeder plus stem cells, although the ANOVA of the three sets of data produced the same statistical conclusions for lysine consumption (*p* < 0.05) and glutamate production (*p* < 0.0001).

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
