# Peer review of "Lysine Deprivation during Maternal Consumption of Low-Protein Diets Could Adversely Affect Early Embryo Development and Health in Adulthood"

_ijerph, 2020, doi:10.3390/ijerph17155462_

Round 1

Reviewer 1 Report

The study showed that LKRSDH mRNA is expressed in hES cell lines, lysine was preferentially consumed by hES cells, and hES cells expresse two glutamate receptors, suggesting that lysine is a  source of glutamate.

Comments:

- There is a lack of details in the material and methods section. For example, information on cell culture conditions, preparation and analysis of microarray, and preparation and analysis of RNA sequencing are very limited.

- Authors should provide protein data.

- In Figure 1, what is the microarray data plus RNA seq data? Merging data from microarray and RNA seq analyses in a graph is considered to be acceptable?

- In Figure 1, what is the rationale for comparing expression of LKRSDH mRNA with other genes? This reviewer does not agree with the idea that a higher expression of LKRSDH as compared with housekeeping genes is biologically important. Determining change in gene expression as ES cells differentiate seems to be more important.

Author Response

- There is a lack of details in the material and methods section. For example, information on cell culture conditions, preparation and analysis of microarray, and preparation and analysis of RNA sequencing are very limited.

Additional details are provided in the Materials and Methods section as shown using track changes.

- Authors should provide protein data.

Since additional experiments are not possible, would it be better to change our paper from an “Article” to a “Communication”?

- In Figure 1, what is the microarray data plus RNA seq data? Merging data from microarray and RNA seq analyses in a graph is considered to be acceptable?

Remarkably, the expression pattern was about the same regardless of the instrument we used to acquire the data and was still highly statistically significant when we combined them.

- In Figure 1, what is the rationale for comparing expression of LKRSDH mRNA with other genes? This reviewer does not agree with the idea that a higher expression of LKRSDH as compared with housekeeping genes is biologically important. Determining change in gene expression as ES cells differentiate seems to be more important.

It is believed that the gene expression profile reflects how cells are functioning. For instance, the major active expressing network in hES cells belongs to self-renewal and pluripotency, while many other genes are silent. Since microarray/RNAseq expression data are presented in relative units such as hybridization intensity values, the housekeeping genes are used for the identification of the expression level, such as “low”, “moderate” and “high”. We believe Figure 1 helps to better present the biological context of the discussion.

However, the idea proposed by the reviewer to determine LKRSDH mRNA expression change in differentiated phenotypes is excellent.  We investigated LKRSDH mRNA expression after hES cells differentiated as described in Materials and Methods (page 3), and present these data in Results on page 4 where we state “Moreover, alpha-aminoadipic semialdehyde synthase RNASeq expression decreased to 1/22 of its level in H1 hES cells as the cells differentiated into mesenchymal cells (p < 0.0001), and a similar conclusion applied to osteogenic, vascular, and neural differentiation (ANOVA; N = 12 for hES cells vs. four types of differentiation; p < 0.0001 for hES vs. each type of differentiated cell).”

Reviewer 2 Report

Great work. The manuscript is well written and adequately addresses the research questions. There are some minor editing errors such as spaces in the background that can be corrected during final proofread. Please ensure to do so. Thank you.

Author Response

Great work. The manuscript is well written and adequately addresses the research questions. There are some minor editing errors such as spaces in the background that can be corrected during final proofread. Please ensure to do so. Thank you.

Thank you.  We have performed the proofreading.

Reviewer 3 Report

In this manuscript, Van Winkle et al has performed a nice/interesting study to investigate the role of glutamate production from lysine in regulating pluripotency and proliferation of hES cells. The implication of this study could be very important and would of interest to a large pool of readers. I commend the authors for the introduction of such a nice hypothesis and subsequently testing it in relevant model. However, I would like to see few additional experiments before the concept can be full-proof and be published in IJERPH. If the other reviewers agree with the current format, I would at least like the authors to discuss these topics in the absence of actual experiments:

(1) The authors have measured the RNA levels of the enzyme LKRSDH. How is the protein level of this enzyme in different human tissues? Authors should at least check Human Protein Atlas and include this information.

(2) What is the loss-of-function phenotype of this enzyme and could exogenous glutamate treatment would rescue it? Additionally, if the author’s hypothesis is true, then only glutamate treatment would rescue LKRSDH phenotype, but lysine treatment would not. What do the authors think about it?

(3) Is there any disease reported for LKRSDH mutations?

(4) Finally, I think it is very important for the authors to clarify the importance of their study. Do the authors want to propose (a) glutamate is important for hES cells or (b) LKRSDH-mediated conversion of lysine to glutamate is critical and is used as a source of glutamate. The way authors start writing their story (specially in abstract), it seems like they are approaching to answer second possibility. However, after reading their complete results, it is clear that although they have provideded some evidence that glutamate is important for hES cells, but I am afraid to say, there is no evidence that conversion of glutamate from lysine is critical. In my opinion, the RNA expression data for LKRSDH is the only evidence, but that is a very weak link since it could be completely unrelated. Hence, I think that the authors either need to show this link following strategies similar to mine in point 2 or need to do re-write the focus of this paper.   

Author Response

In this manuscript, Van Winkle et al has performed a nice/interesting study to investigate the role of glutamate production from lysine in regulating pluripotency and proliferation of hES cells. The implication of this study could be very important and would of interest to a large pool of readers. I commend the authors for the introduction of such a nice hypothesis and subsequently testing it in relevant model. However, I would like to see few additional experiments before the concept can be full-proof and be published in IJERPH. If the other reviewers agree with the current format, I would at least like the authors to discuss these topics in the absence of actual experiments:

Since additional experiments are not possible, would it be better to change our paper from an “Article” to a “Communication”?

(1) The authors have measured the RNA levels of the enzyme LKRSDH. How is the protein level of this enzyme in different human tissues? Authors should at least check Human Protein Atlas and include this information.

We now state at the beginning of our discussion “In the major pathway for its catabolism, lysine is first converted to alpha-aminoadipic semialdehyde and glutamate by alpha-aminoadipic semialdehyde synthase [14-16].  Human fibroblasts, astrocytes, and neural progenitor cells each express this protein [16].” 

The studies cited [i.e., 14-16] not only show expression of the protein and its RNA, but, more importantly, they demonstrate actual enzyme activity.

(2) What is the loss-of-function phenotype of this enzyme and could exogenous glutamate treatment would rescue it? Additionally, if the author’s hypothesis is true, then only glutamate treatment would rescue LKRSDH phenotype, but lysine treatment would not. What do the authors think about it?

As we also now state near the beginning of our Discussion “The pool of glutamate produced in this way must be unique in comparison to glutamate formation from other amino acids.  Human alpha-aminoadipic semialdehyde synthase deficiencies are associated with severe physiological and morphological defects in the nervous system [14].”

That is, glutamate produced from lysine must form a unique glutamate pool for which other sources of glutamate cannot completely compensate or rescue.  In addition, we predict that alpha-aminoadipic semialdehyde synthase deficiencies might be associated with metabolic syndrome and related adult disorders (owing to an effect of the deficiencies on stem cell progenitors in the blastocyst), although such studies have not, to our knowledge been performed.

(3) Is there any disease reported for LKRSDH mutations?

Yes, please see responses to point 2 above.

(4) Finally, I think it is very important for the authors to clarify the importance of their study. Do the authors want to propose (a) glutamate is important for hES cells or (b) LKRSDH-mediated conversion of lysine to glutamate is critical and is used as a source of glutamate. The way authors start writing their story (specially in abstract), it seems like they are approaching to answer second possibility. However, after reading their complete results, it is clear that although they have provideded some evidence that glutamate is important for hES cells, but I am afraid to say, there is no evidence that conversion of glutamate from lysine is critical. In my opinion, the RNA expression data for LKRSDH is the only evidence, but that is a very weak link since it could be completely unrelated. Hence, I think that the authors either need to show this link following strategies similar to mine in point 2 or need to do re-write the focus of this paper.  

Please see our reply to point 2 above.

Round 2

Reviewer 1 Report

The authors have addressed my major concerns, and the manuscript has been significantly improved but have not shown the new data. The data showing LKRSDH mRNA expression after hES cells differentiated needs to be added in Figure 1.

Author Response

LKRSDH mRNA expression after hES cell differentiation into four types of cells has been added to Figure 1.